# Improved Optimal Control of Transient Power Sharing in Microgrid Using H-Infinity Controller with Artificial Bee Colony Algorithm

Mohammed Said Jouda *  and Nihan Kahraman *

Department of Electronics and Communication Engineering, Yıldız Technical University, Esenler, Istanbul 34220, Turkey
* Correspondence: m.s.jouda@gmail.com (M.S.J.); nicoskun@yildiz.edu.tr (N.K.)

**Abstract:** The microgrid has two main steady-state modes: grid-connected mode and islanded mode. The microgrid needs a high-performance controller to reduce the overshoot value that affects the efficiency of the network. However, the high voltage value causes the inverter to stop. Thus, an improved power-sharing response to the transfer between these two modes must be insured. More important points to study in a microgrid are the current sharing and power (active or reactive) sharing, besides the match percentage of power sharing among parallel inverters and the overshoot of both active and reactive power. This article aims to optimize the power response in addition to voltage and frequency stability, in order to make this network's performance more robust against external disturbance. This can be achieved through a self-tuning control method using an optimization algorithm. Here, the optimized droop control is provided by the H-infinity (H∞) method improved with the artificial bee colony algorithm. To verify the results, it was compared with different algorithms such as conventional droop control, conventional particle swarm optimization, and artificial bee colony algorithms. The implementation of the optimization algorithm is explained using the time domain MATLAB/SIMULINK simulation model.

**Keywords:** microgrid; optimization; power sharing; droop control; artificial bee colony algorithm; particle swarm optimization; H∞ optimal controller; ABC



## 1. Introduction

Microgrid technology has brought huge flexibility in power and operating system control. The network consists of several parallel distributed generators (DGs) with control techniques [1] to investigate the power stability and robustness against any disturbance. On the other hand, a microgrid network may involve renewable energy sources such as photo-voltaic, wind-turbine, micro-turbine systems [2,3]. Figure 1 illustrates the structure of the microgrid network, where energy storage systems (ESS) with other sources are connected to the common AC bus and distributes the power to the loads. At a common coupling point (PCC), the power flows between the main grid and microgrids to share the power [4,5]. In order to increase the efficiency of the microgrids, the power inverters must have a high control performance. The method of droop control is designed to investigate the active and reactive power sharing without any communication protocol to ensure the voltage and frequency stability of the microgrid with a reduced deviation value between the active power–frequency (P–f) and reactive power–voltage (Q–V) under inductive impedance condition [6,7]. This paper improves the performance of power-sharing between parallel inverters by developing the droop control method with advanced level optimal control using an optimization algorithm. Some improved control methods have already been developed by researchers. For example, the droop control performance improved with high resistive transmission lines, and virtual impedance is discussed in [8,9].

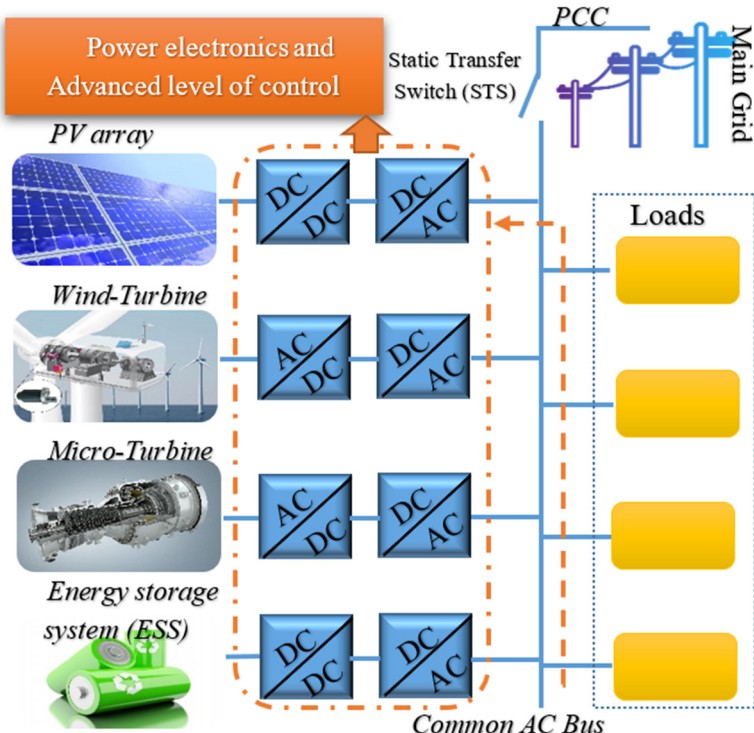

**Figure 1.** Structure of the microgrid.

There are many problems solved through the adaptive PI controller to eliminate the current sharing error, and an enhanced control technique with droop control is presented in [10,11]. Obtaining efficient power sharing dynamic performance under the complex impedance line condition by the droop control method is discussed in [12]. The power management technique was applied in the microgrid by increasing the virtual initial to control the energy storage systems' transient response [13]. The robust control technique was used in island-mode for voltage and frequency regulation to compensate the power-sharing between parallel inverter sources by a circular limited cycle oscillator frequency locked loop with pre-filter (CLO-FLLWPF) control [14]. The droop control of virtual-impedance using capacitive-coupling inverters to minimize the power sharing error [15]. Many recent studies have covered improving the network quality and the management of microgrids. For instance, Harris hawks optimization (HHO), and the water cycle algorithm (WCA) are presented in [16,17]. Tuning the value of the voltage and frequency according to the rated values by the hierarchal secondary control is presented in [18]. The disadvantages of many control techniques such as conventional droop control are slow dynamic response, deviation in the voltage and frequency, a mismatch, and poor power-sharing between the DGs in the cases of nonlinear or unbalanced loads. In addition, VPD/FQB droop control has poor voltage regulation and frequency regulation, angle droop control has poor power-sharing, and the single injection method has a distortion in the harmonic voltage [19]. In order to solve these problems, droop control should be improved to satisfy the best results in power sharing.

This paper proposes an optimized droop control by the H∞ method with the artificial bee colony algorithm (ABC). The paper investigates the optimal transient power-sharing between the parallel inverters through the island mode. The controller was designed in island mode using a small-signal model of a microgrid consisting of two inverters. The paper also proposes the new droop arrangement to enhance the active and reactive power-sharing without requiring continuous monitoring of the point of common coupling (PCC) voltage. We achieved the stability between the DC link and inverter power flow by applying the H∞ PID controller with ABC droop control to decide the setpoint power of the active and reactive reference power of AC inverters, besides testing the proposed controller under

various load situations and comparing it to another controller's performance. This research focused on achieving the following points: (1) eliminate the steady-state errors for the output signals; (2) decrease the overshoot of transient response as appropriate for the application and the response will be faster to reach a steady state mode when transferred between the grid-connected and islanded mode, thus minimizing the overshoot response and fast dynamic response; (3) improving robustness against external disturbance refers to errors that affect the stability of the microgrid such as voltage drop and frequency deviation; (4) a perfect match for inverters besides improving the power-sharing between parallel distributed generators; and (5) a DC link voltage with control algorithms. Analytical methods of droop control to design the self-tuning PID of the coefficient power by H∞ optimal controller using the artificial bee colony (ABC) algorithm is demonstrated and the parallel inverters stability performance is discussed, in another side, the results with ABC algorithm without using H∞ controller is compared to conventional droop control and particle swarm optimization (PSO) algorithm.

The rest of this article is organized as follows. Section 2 discusses the mathematical modeling and optimization algorithm, particle swarm optimization algorithm, and artificial bee colony algorithm. Section 3 displays the controller design with an H∞ optimal controller with ABC. Optimized droop control parameters by the H∞ optimal controller with the ABC algorithm is shown in Section 4. DC link voltage and circulating power of the inverters are illustrated in Section 5. The results are presented in Section 6. The conclusions are presented in Section 7.

## 2. Mathematical Modeling and Optimization Algorithms

This section describes the optimization algorithm that applies to parallel inverters to achieve active and reactive power sharing and minimize the harmonic current in the parallel inverters. The analyses concluded that conventional droop control cannot obtain power-sharing for the nonlinear load case. The microgrid consists of several parallel inverters, and the general structure of the microgrid is illustrated in Figure 1. The suggested design of two parallel inverters is shown in Figure 2, where $P^*$ and $Q^*$ are the active and reactive power nominal values, and $P$ and $Q$ are the active and reactive power actual values.

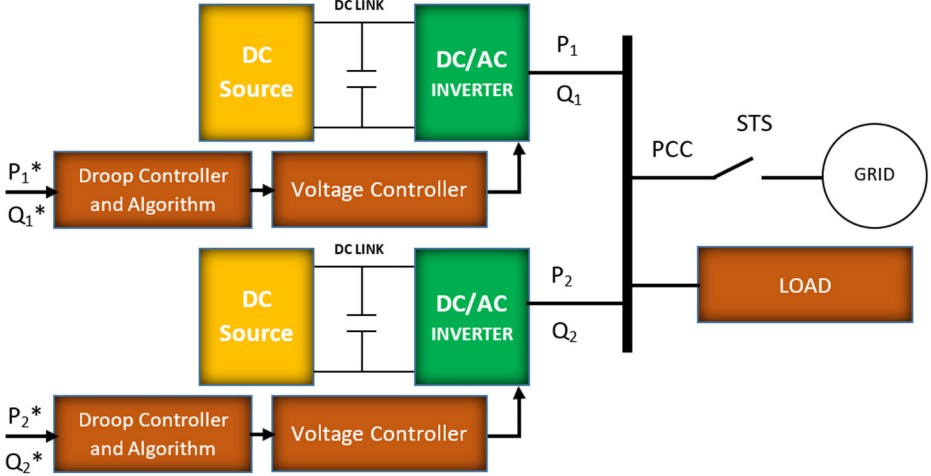

**Figure 2.** Two inverters in the microgrid.

The two inverters were connected as parallel in the microgrid. $V_1$ and $V_2$ are the output voltage of the inverters. $V_L$ is the voltage of the load. $I_{O1}$ and $I_{O2}$ are the output current of the inverters. $\varphi_1$ and $\varphi_2$ are the voltage angle. $\theta_1$ and $\theta_2$ are the impedance angle. $L_1$ and $L_2$ are the line impedance. Lf1 and Cf1 are the filter 1 parameters. Lf2 and Cf2 are the filter 2 parameters. The equivalent block diagram of two inverters in the microgrid are illustrated in Figure 3.

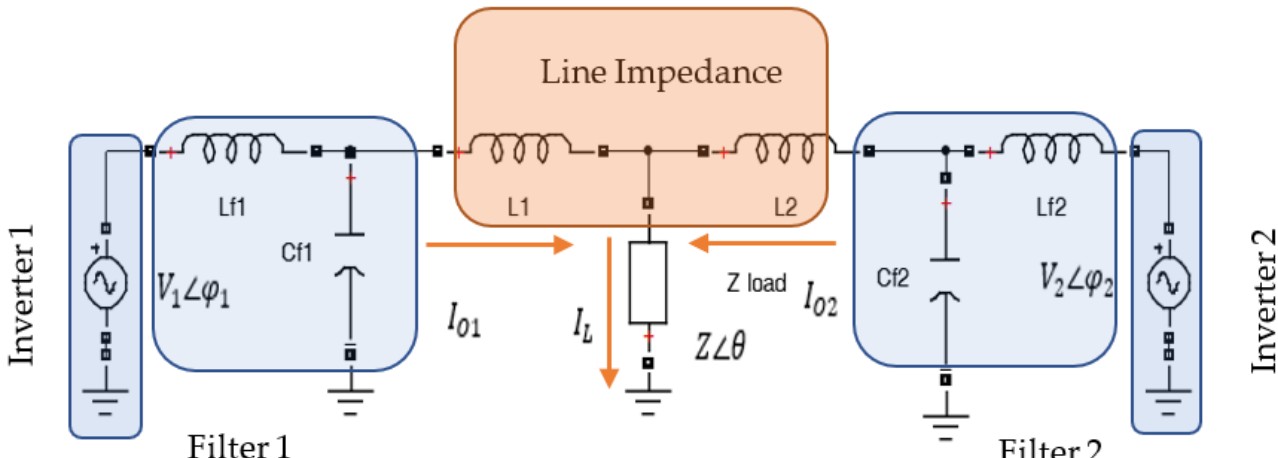

**Figure 3.** Equivalent block diagram of two inverters in the microgrid.

The schematic diagram of the microgrid is shown in Figure 4. The structure of the microgrid can be designed as two inverters. The DC link must connect in the input of the inverters between the DC supply and the IGBTs to keep the inverters working without interruption, besides the diodes that protect the DC source from the reverse current, in addition to connecting the LCL filter in the inverter's output side and measuring the output voltage and current to extract the real and reactive power. The switches in the input and output side were employed to make the isolation of the inverters from the DC power supply and the main grid, respectively. The controller unit was used for the implementation of the optimization algorithm by the time domain MATLAB/SIMULINK. All signals from sensors and measurement values were also processed in the controller module to control the switches and the output signals sent to the inverters. All data, results, and response signals can be displayed.

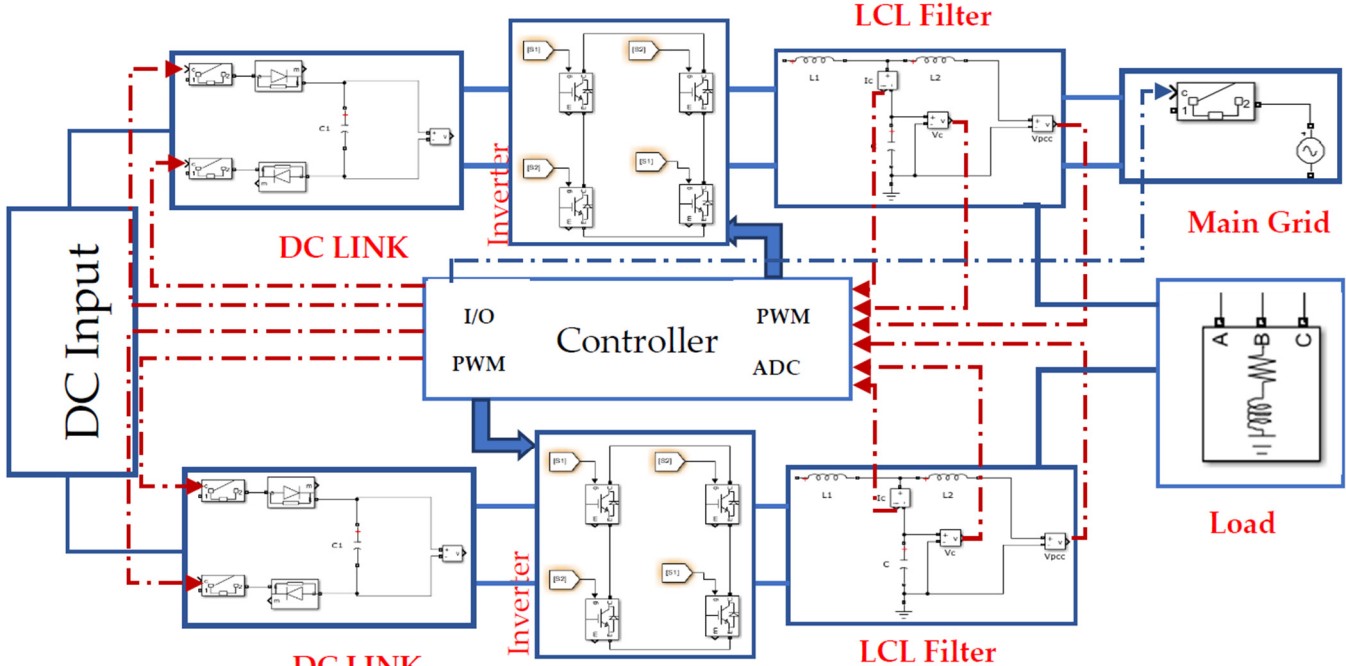

**Figure 4.** Schematic diagram of the microgrid.

### 2.1. Particle Swarm Optimization Algorithm

The PSO algorithm is one of the methods used for robust and stable control. It consists of some particles that are randomly chosen. The representation of these particles' position can be calculated according to the following equations as shown in (1) and (2):

$$V_i^{K+1} = W^k\ V_i^K + C_1\ r_1 \left[ P_{best,i} - X_i^{k+1} \right] + C_2\ r_2 \left[ P_{gbest,i} - X_i^{k+1} \right] \tag{1}$$

$$X_i^{k+1} = X_i^k + V_i^{K+1} \tag{2}$$

The $i$ and $k$ are the particle index and iteration index, respectively. $C_1$ and $C_2$ are weighting factors. $r_1$ and $r_2$ are random numbers between {0,1}. W is a weighting function; $V_i^K$ is the *ith* particle velocity at *kth* iteration. $X_i^k$ is the particle's current position. $P_{best}$ and $P_{gbest,i}$ are the particle's best position and global group's global best position, respectively [20].

The particle velocity and the new position of particles can be updated by applying Equations (1) and (2) to obtain the best particle position group. The main goal is to improve the conventional droop control method by using the PSO algorithm to achieve good load sharing and transient response, as shown in Figure 5 [21]. The results are compared to the H∞ controller with the ABC algorithm, as explained in Section 4.2.

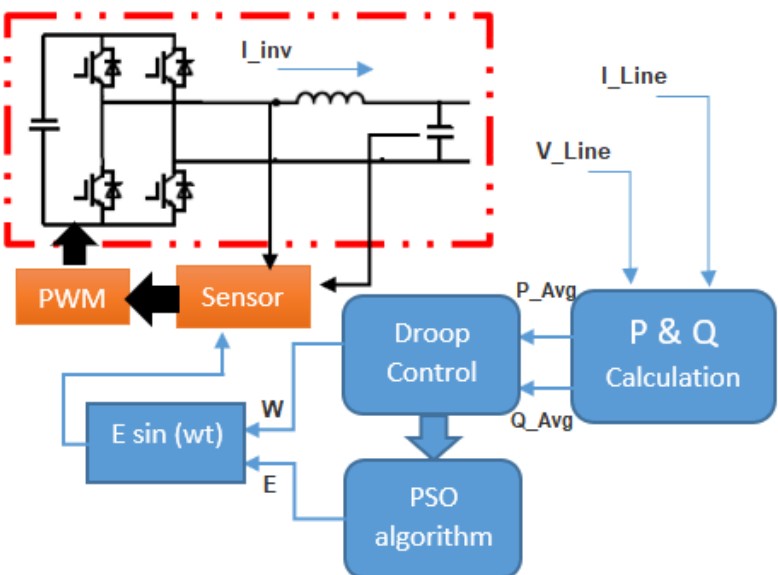

**Figure 5.** Scheme of the PSO droop control method.

### 2.2. Artificial Bee Colony Algorithm

This algorithm is one of the best algorithms for optimization problems. It is analyzed as follows: (1) analysis of the real problem; (2) algorithm and the solution technique by numerical methods; (3) computer implementation; and (4) verification and validation of the results. The algorithm can be simulated by bee behavior as it consists of some groups, which are food sources, employed, onlookers, and scout bees. The food sources are very important items to determine the possible solution for the problems on the information on the nectar amount. The employed bees can determine the new solution. The scouts search around the nest for a new food source, while the onlookers work for food creation, based on information sent by the employed bees [22]. The mathematical equation of this algorithm is used as shown in Equation (3):

$$V_{ij} = X_{ij} + \varphi_{ij} \left( X_{ij} - X_{kj} \right) \tag{3}$$

The $V_{ij}$ is a new food source (*i*, *j* are random selected index), and k = {1, 2, . . . . . . , colony size (population)}; $X_{ij}$ is the selected random food source; and $\varphi_{ij}$ is a random number {−1, 1}. After calculating the source position $V_{ij}$, and then verifying these values, they are compared with old values $X_{ij}$. If the new source of food is better than the old source, it will be replaced with that old value. In this algorithm, there are two important parameters in which to apply this procedure. These parameters directly affect the algorithm performance, namely, colony size (population) and the maximum number of iterations.

Analytically, the optimum values of the overshoot, settling time, and power-sharing between the parallel inverters were achieved by the ABC algorithm with droop control when compared to other algorithms such as the conventional PID and PSO algorithms. It will be clear that the best values were obtained, as shown in Table 1. The best parameters of PID control of the state-space equation that have minimum overshoot, and minimum settling time when the ABC algorithm is used and investigates the optimal transient power-sharing between the parallel inverters through the island mode, as shown in Sections 6.1 and 6.2.

**Table 1.** Step response characteristics of different algorithms.

| Method | Control PID Parameters | Step Response Characteristics | | |
|---|---|---|---|---|
| | | OS % | Ts (S) | Absolute Error |
| **Conventional** | Kp = 0.64 Ki = 0.24 Kd = 0.308 | 58.2 | 23.3 | 25.8 |
| **PSO** | Kp = 0.9005 Kd = 0.5284 | 15.8 | 4.18 | 15.6004 |
| **ABC** | Kp = 0.5885 Kd = 0.1988 | 7.41 | 7.24 | 14.2117 |
| **ABC with H∞** | Kp = 0.313 Kd = 0.225 | 1.26 | 7 | 10.529 |

The ABC algorithm does not need more iteration, since it works in fewer repetitions. Therefore, this algorithm needs far fewer variables than other algorithms. The most important feature of the ABC method is that the parameters of PID tuning can easily be obtained according to a fitness value. Normally, the ABC algorithm needs the most calculation to achieve the optimum solution [23,24], but it can obtain high improvements with low iterations, therefore, it is preferable to use this method to obtain the best results with the least number of iterations. Moreover, the performance of step response depends on the algorithm, and the PID parameters can be adjusted through them. Finally, this article is motivated by the ABC algorithm with droop control to solve the optimization problem to minimize the over-shoot, decrease the settling time, and minimize the feedback absolute error. According to the results, this algorithm can achieve a high-quality solution. In addition, the control parameters can be modified easily by the ABC algorithm.

### 3. Controller Design

#### 3.1. Design of H∞ Optimal Controller

The H∞ optimal controller is used to minimize the worst-case gain of the control system. This method can be used for applications that need to minimize the error cost function. The dynamic objective function is modeled by a generic state equation, as in Equation (4):

$$x(t) = Ax(t) + \beta_u u(t) + \beta_w w(t) \tag{4}$$

The function of the H∞ optimization problem depends on multiple inputs $f(x(t), u(t), w(t))$, while $x(t)$ is called the state of the system, and the system matrix, A. The differential equation consists of the optimal control input $u(t)$. Let the system external input (i.e., a worse-case disturbance input $w(t)$) and the input matrix be Bu and $\beta_w$. Figure 6 illustrates the construction of the H∞ optimal full information control to

minimize the objective function $F = \min(\max f[x(t), u(t), w(t)])$. In the feedback system, after calculating the infinity norm, the disturbance inputs can be predicted. This part demonstrates a multiple input multiple output (MIMO) design and compares it with a single input single output (SISO) to improve the responsiveness and reduce the external disturbances. Figure 7 shows the H∞ optimal controller of the closed-loop system with plant state, disturbance input, reference output, and feedback system.

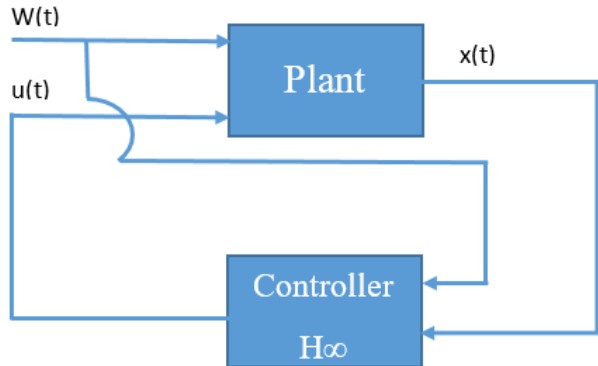

**Figure 6.** H∞ controller with full control.

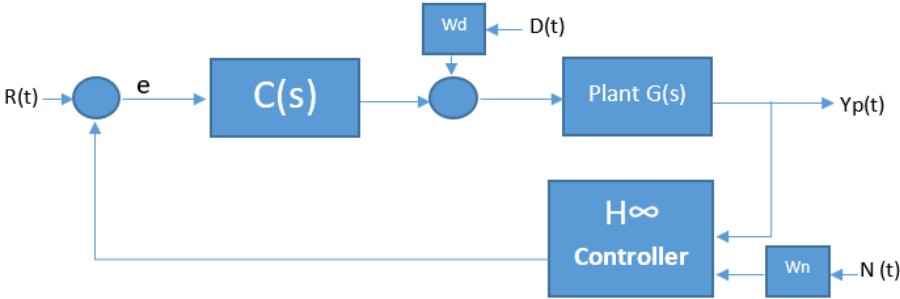

**Figure 7.** The closed-loop system with the H∞ controller.

The closed-loop system consists of an uncertain plant $G(s)$ that has an n-input and m-output. The output system is represented as Yp(t), R(t) is a reference input, e(t) is an error signal of the closed-loop control system, $C(s)$ is the tuning algorithm of the PID controller, $W_d$ and $W_n$ are weighting functions, and D(t) and N(t) are the disturbance and the noise, respectively [25]. The amount of disturbance and the noise can be determined through the infinity norm, as shown in Equation (5):

$$\left\| G(s)(I + C(s)G(s))^{-1} W_d, \ (I + C(s)G(s))^{-1} W_n \right\|_\infty \tag{5}$$

The H∞ optimal controller is used to design a strong performance for robust closed-loop systems to reduce uncertainty. High-order systems are difficult to implement practically, so the high-order can be reduced by the H∞ control technique [26,27]. Therefore, several problems in industrial applications have been solved by the PID controller. As a result, it needs to develop a more stable system to achieve efficient power-sharing with a droop controller using PID H∞ with the ABC algorithm.

### 3.2. Design of PID H∞ Optimal Controller with ABC Algorithm

This article proposes a closed-loop control system designed by the droop controller using an H∞ optimal controller with the ABC algorithm to eliminate plant uncertainty, as shown in Figure 8.

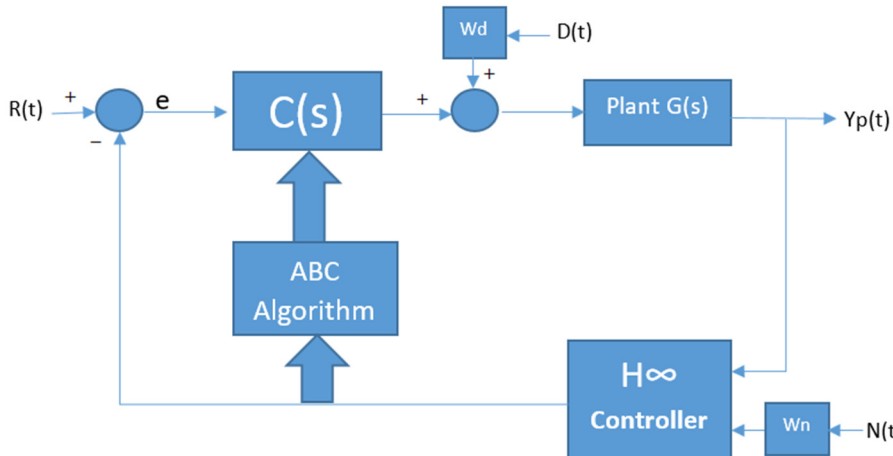

**Figure 8.** A closed-loop system with the PID H∞ controller using the ABC algorithm.

*n* is the number of inputs, *m* is the number of outputs, D(t) is an external disturbance, N(t) is a noise signal, R(t) is a reference input, e(t) is an error signal of the closed-loop control system, and Yp(t) is the output system. The tuning algorithm of the PID controller *C(s)* can be presented as in Equation (6).

$$
C(s) = \begin{bmatrix} K_{P_{11}} & \cdots & K_{P_{1n}} \\ \vdots & \ddots & \vdots \\ K_{P_{m1}} & \cdots & K_{P_{mn}} \end{bmatrix} + \frac{1}{S} * \begin{bmatrix} K_{I_{11}} & \cdots & K_{I_{1n}} \\ \vdots & \ddots & \vdots \\ K_{I_{m1}} & \cdots & K_{I_{mn}} \end{bmatrix} + S * \begin{bmatrix} K_{D_{11}} & \cdots & K_{D_{1n}} \\ \vdots & \ddots & \vdots \\ K_{D_{m1}} & \cdots & K_{D_{mn}} \end{bmatrix} \tag{6}
$$

The stability of the system depends on the decrease in the disturbance attenuation to achieve the weighting coefficient that affects the objective function, where the equations that will give the *T(s)* and *D(s)* are the complementary function and sensitivity function, respectively, as shown in Equations (7) and (8).

$$
T(s) = \left[ C(s){\cdot}G(s) * (I + C(s){\cdot}G(s))^{-1} \right] \tag{7}
$$

$$
D(s) = [I + C(s){\cdot}G(s)]^{-1} \tag{8}
$$

The robust and disturbance attenuation performance can be identified by a weighting function matrix (*W1*, *W2*) according to Equations (7) and (8), as shown in Equations (9) and (10), respectively:

$$
P_1 = ||W1 * T(s)||_\infty \leq 1 \tag{9}
$$

$$
P_2 = ||W2 * D(s)||_\infty \leq 1 \tag{10}
$$

Equations (9) and (10) are combined to satisfy the infinity-optimized controller design as $P_\infty = \sqrt{P_1^2 + P_2^2}$.

There are several types of performance criteria in the controller design aspect such as the integral of absolute error (*IAE*), the integral of squared-error (*ISE*), and the integral of time-weighted-squared-error (*ITSE*). It has been shown in [28] that the performance criterion of the *ITSE* is better than the *IAE* and *ISE*, on one hand, to overcome the maximum peak value of overshoot. Equations for the performance criteria are illustrated in Equations (11)–(13):

$$
IAE = \int_0^\infty | \, Input(t) - Output(t)|dt = \int_0^\infty |e(t)|dt \tag{11}
$$

$$
ISE = \int_0^\infty e^2(t) \, dt \tag{12}
$$

$$ITSE = \int_0^\infty te^2(t)\,dt \tag{13}$$

where $e(t)$ is the error signal that can be calculated by subtracting the power value of the parallel inverter from the power nominal value, and $t$ refers to the simulation time. Furthermore, the performance of the controller design can be expressed as shown in Equation (14):

$$I(\theta) = a_1 P_\infty + a_2 OS + a_2 t_r + a_3 t_s + a_4 E_{ss} + a_5 \int_0^T te^2(t)dt \tag{14}$$

where $P\infty$ is the robust stability performed by H∞ optimization; $OS$ is overshoot; $t_r$ is rising time; $t_s$ is settling time; $E_{SS}$ is steady-state error; $a_i$ ($i = 1, 2, \ldots, 5$) is the weight factor; $I(\theta)$ is the performance for weighting function; and $\theta$ is the coefficient controller. For single input single output (SISO), $\theta$ is expressed by a vector (Equation (15)):

$$\theta = [K_p \; K_I \; K_D] \tag{15}$$

where $K_p$, $K_I$, and $K_D$ are proportional, integral, and derivative controllers, respectively. For multiple-input multiple-output (MIMO), expressed by a matrix ($n$-input $\times$ $m$-output) (Equation (16)):

$$\begin{aligned}\theta = \big[ & K_{P_{11}} \ldots \; K_{P_{1n}} \; K_{P_{21}} \ldots \; K_{P_{2n}} \ldots \cdot \ldots K_{P_{m1}} \ldots K_{P_{mn}} \\ & K_{I_{11}} \ldots K_{PI_{1n}} K_{I_{21}} \ldots K_{I_{2n}} \ldots \cdot \ldots K_{I_{m1}} \ldots K_{I_{mn}} \\ & K_{D_{11}} \ldots K_{D_{1n}} K_D \ldots K_{D_{2n}} \ldots \cdot \ldots K_{D_{m1}} \ldots K_{D_{mn}} \big] \end{aligned} \tag{16}$$

## 4. Optimized Droop Control Parameters by H∞ Optimal Controller with ABC Algorithm

The droop control method using the H∞ optimal controller with the ABC algorithm that decides the power coefficient compensator values for best power-sharing among parallel inverters is shown in Figure 9. The following equations used to extract the algorithm parameters, the reference input, and the external disturbance of the system D(t) as well as determine the uncertainty weight Wd (s), a PD structure, and the individual control parameters in ABC algorithm are represented in Equations (17) and (18), respectively:

$$\text{Compensator} = \begin{bmatrix} K_{P_{11}} & \cdots & K_{P_{1n}} \\ \vdots & \ddots & \vdots \\ K_{P_{m1}} & \cdots & K_{P_{mn}} \end{bmatrix} + S \begin{bmatrix} K_{D_{11}} & \cdots & K_{D_{1n}} \\ \vdots & \ddots & \vdots \\ K_{D_{m1}} & \cdots & K_{D_{mn}} \end{bmatrix} \tag{17}$$

$$\begin{aligned}\theta = \big[ & K_{P_{11}} \ldots K_{P_{1n}} \; K_{P_{21}} \ldots K_{P_{2n}} \ldots \ldots K_{P_{m1}} \ldots K_{P_{mn}} \\ & K_{D_{11}} \ldots K_{D_{1n}} \; K_D \ldots K_{D_{2n}} \ldots \ldots K_{D_{m1}} \ldots K_{D_{mn}} \big] \end{aligned} \tag{18}$$

The frequency and voltage equations of parallel inverters in the microgrid by using ABC droop control are shown in Equations (19) and (20):

$$\omega = \omega^* - (K_p + Kd_p \frac{d}{dt})(P - P^*) \tag{19}$$

$$E = E^* - (K_q + +Kd_q \frac{d}{dt})(Q - Q^*) \tag{20}$$

where $P^*$ and $Q^*$ are the active and reactive power nominal values; $P$ and $Q$ are the active and reactive power actual values; $\omega^*$ and $E^*$ are the frequency and voltage nominal value; $\omega$ and $E$ are the frequency and voltage actual values; and $K_p$ and $K_q$ are the frequency and voltage droop control gain, respectively.

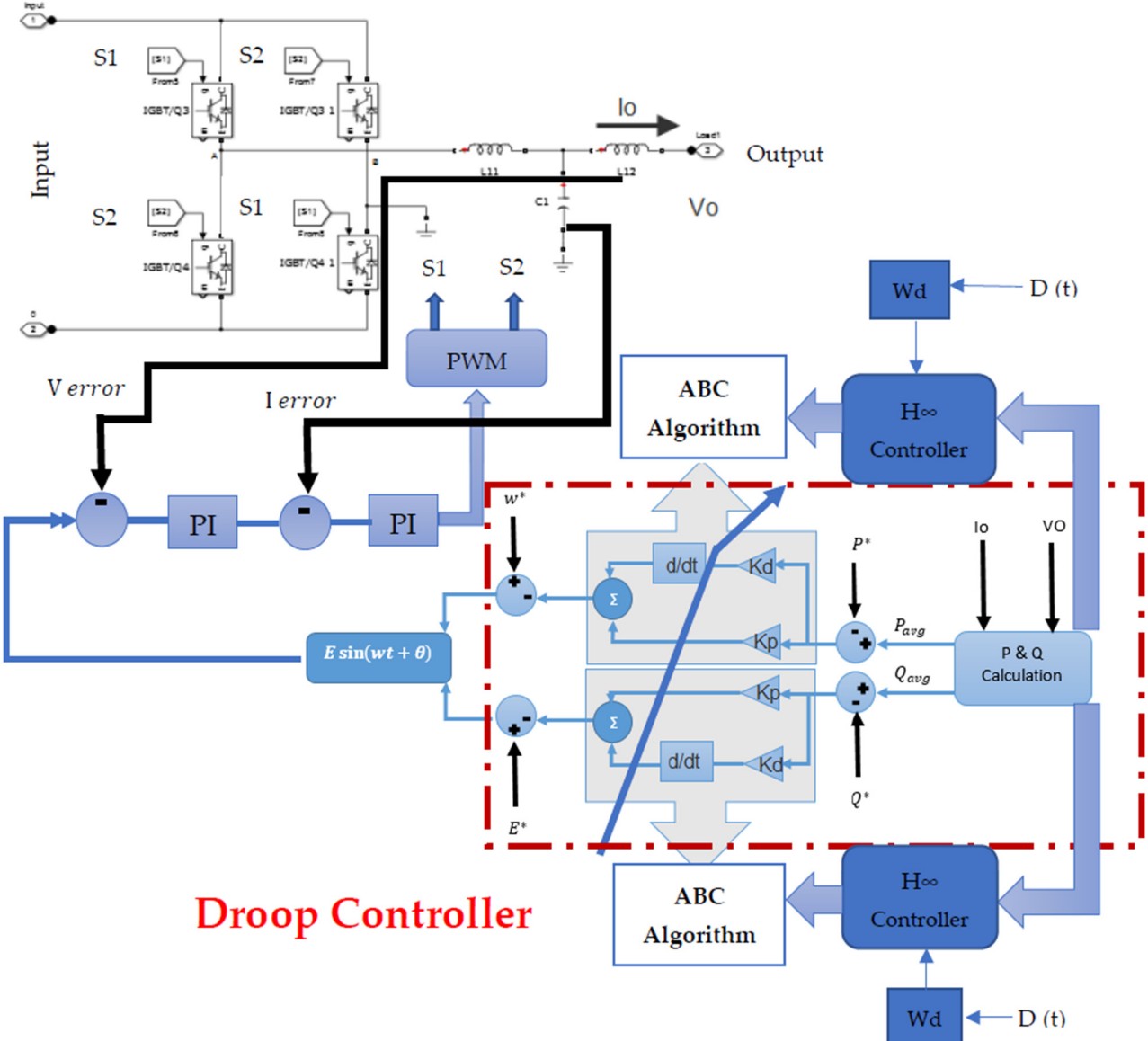

**Figure 9.** Scheme of the parallel inverter module by H∞ optimal droop control with the ABC algorithm.

### 4.1. Explanation Method of H∞ Controller Using ABC

The main goal of the H∞ optimal controller with the ABC algorithm is to find the optimal value of $\theta$ that minimizes the objective function, and it chooses the best coefficient power controller that improves the sharing of active and reactive power among the parallel inverters. The methodology can be described as follows:

- Randomly select the food source for each individual $\theta_k$, where k is the colony size of the ABC algorithm k = {1, 2, . . . . . . , colony size (population)}.
- Calculate the values of the evaluation function in Equation (14) as previously explained, and apply the result in the droop control equations.
- Calculate the new value of the source position $V_{ij}$, where *i* and *j* are randomly selected indexes.
- Compare the new value of the source position of each individual $\theta_k$ with old values according to Equation (3).
- In case the new source food is better than the old source, modify the source food of each individual $\theta_k$ according to the mathematical equation of the ABC algorithm. If not, the new source is the best value of source food.

- Test the maximum number of iterations; if the number of iterations reaches the maximum, this means that the last value is the optimal controller parameter. If not, repeat the process to evaluate the objective function for $\theta_k$, where k = {1, 2 . . . ..., colony size}. The algorithm flowchart is shown in Figure 10.

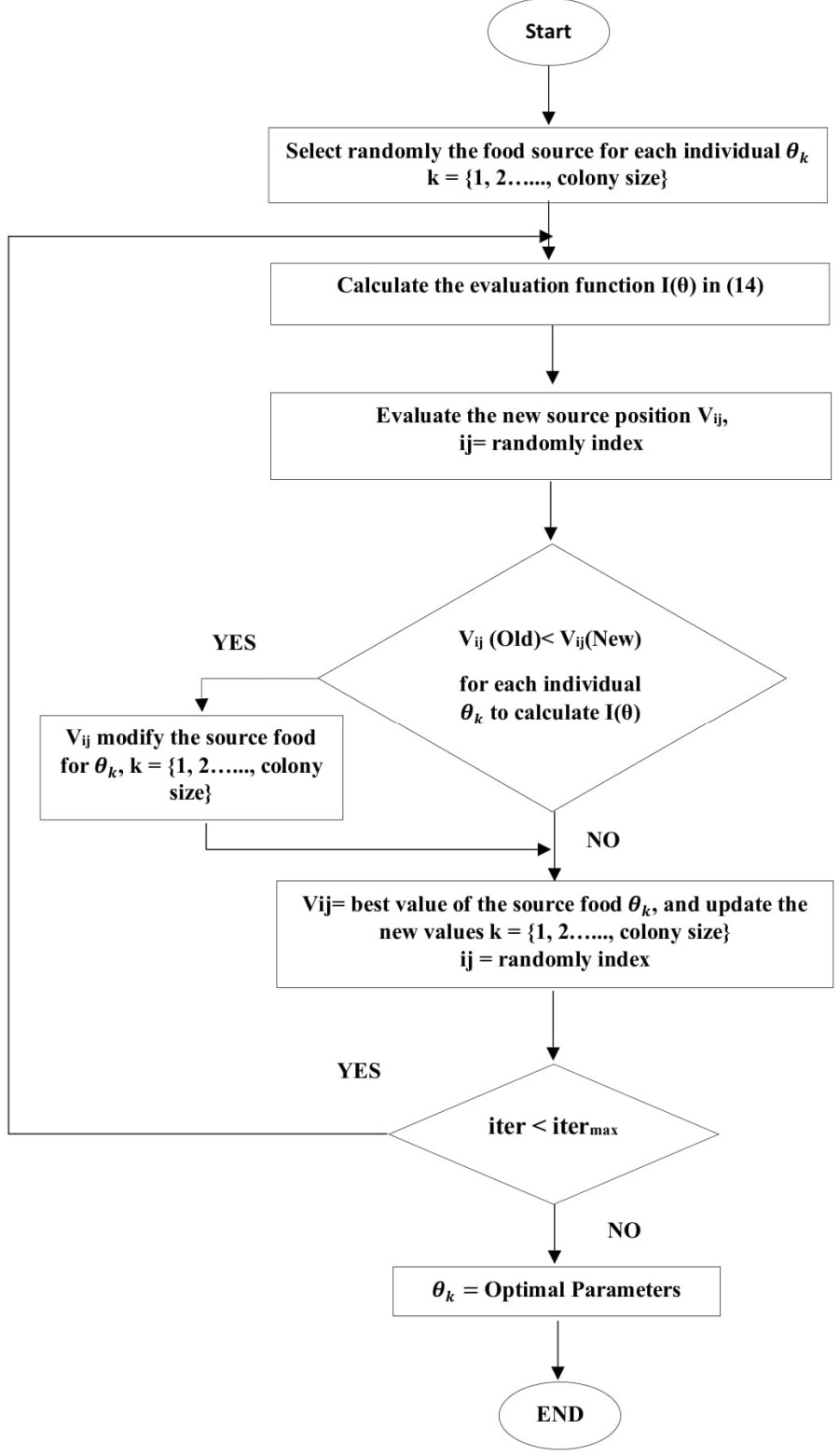

**Figure 10.** Flowchart of the algorithm.

### 4.2. Inverter's Power Regulator

The droop controller was designed to achieve accurate power-sharing and stable output voltage with suitable frequency. The small-signal module for the power sharing feedback closed system is shown in Figure 11. The following transfer function is described by Equation (21). The feedback control system was designed by the droop controller PID tuning method with:

1. Conventional controller.
2. Particle swarm optimization (PSO) algorithm.
3. Artificial bee colony algorithm.
4. H∞ optimal controller with the ABC algorithm.

$$G_P(s) = C * G(s)/(1 + C * G(s) * F(s)) \tag{21}$$

where $C$ is the compensator of the system (droop controller and algorithm); plant $G(s)$ has $n$-inputs and $m$-outputs; and $F(s)$ is the feedback system. The PID controller coefficients play a role in influencing the response, so the values of the maximum overshoot, settling time, and the feedback absolute error to decide the method that achieves optimum control parameters must be studied. In this work, better performances for the controlled system were achieved by PSO, ABC, and the H∞ optimal controller with the ABC algorithm to overcome the problems of the conventional methods. The simulation results are shown in Figure 12 and Table 1, where they show that the algorithms were successful in achieving the desired values of the PID parameters. It can be seen that the conventional controller method is an inaccurate way to obtain the desired goals. The PSO algorithm needs more parameters that use it to store and update the variables; on the other hand, it requires more time to calculate the control system parameters and several iterations that are used in the optimization technique to reach the optimum solution. The ABC algorithm does not need more iteration, since it works in fewer repetitions, this algorithm needs far fewer variables than the PSO algorithm, and it makes this method more desirable for the user than previous methods. The robustness of the step response depends on the choice of the algorithm, as shown in the results in Table 1 according to Figure 11 and the case study parameters in Table 2, where the H∞ controller with the ABC algorithm succeeded in minimizing the overshoot, decreasing the settling time, and also minimizing the error cost function. According to the simulation results, this algorithm achieved the best solution to the control problem. Thus, the controller parameters can improve power-sharing and eliminate the deviation values of voltage in the microgrid by using this H∞ controller with the ABC algorithm, which will be discussed in the next section in detail.

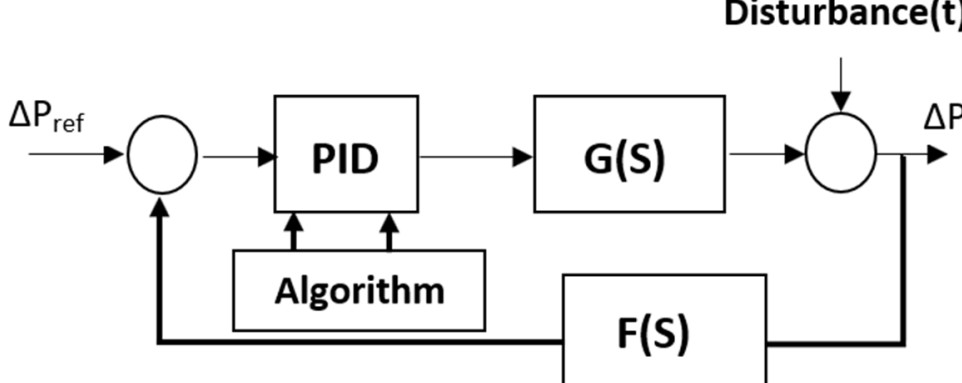

**Figure 11.** Small signal module feedback control system.

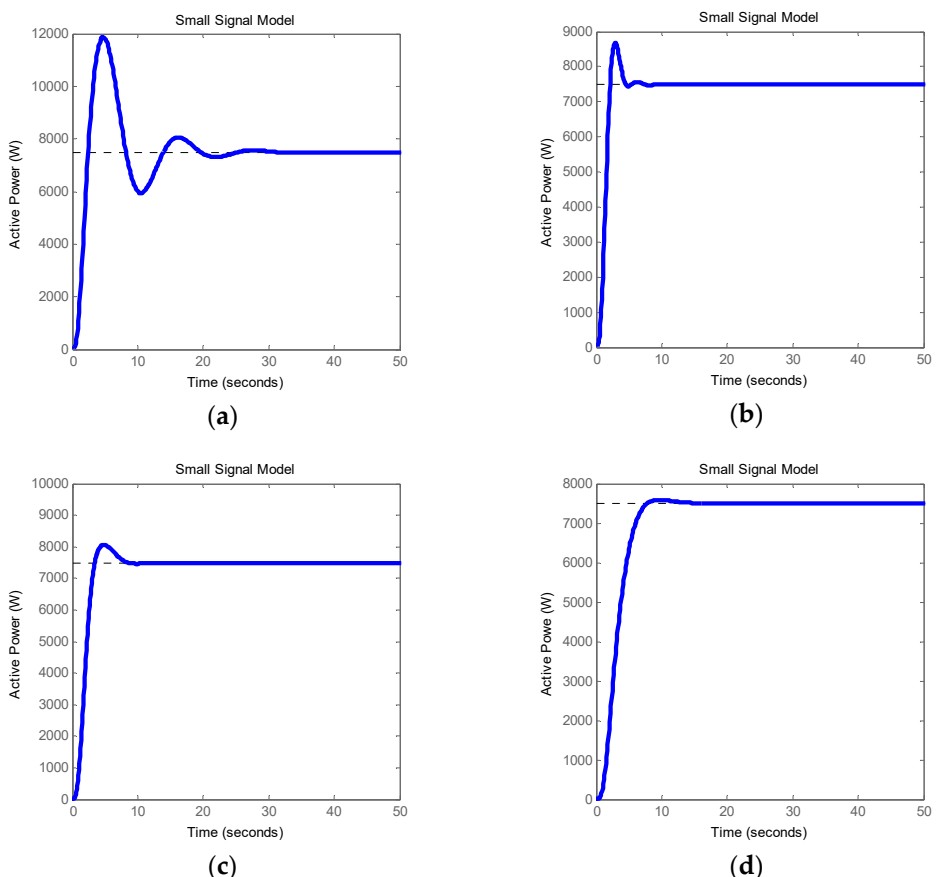

**Figure 12.** Small signal model for the power control system of different algorithms and droop controller by (**a**) conventional PID (**b**) PSO algorithm, (**c**) ABC algorithm, and (**d**) H∞ controller with ABC algorithm.

**Table 2.** Case study parameters.

| Symbol | Description | Value |
|:------:|:-----------:|:-----:|
| $P_1^*$ | Nominal Value of Active Power for inverter 1 | 10 KW |
| $P_2^*$ | Nominal Value of Active Power for inverter 2 | 10 KW |
| $Q_1^*$ | Nominal Value of Reactive Power for inverter 1 | 150 Var |
| $Q_2^*$ | Nominal Value of Reactive Power for inverter 2 | 150 Var |
| $P_L$ | Load Active Power | 7500 W |
| $Q_L$ | Load Reactive Power | 300 Var |
| $E^*$ | Nominal Voltage | 325 V |
| $f^*$ | Nominal frequency | 50 HZ |
| $V_{DC\_Link}$ | DC link voltage | 400 V |

## 5. DC Link Voltage and Circulating Power of the Inverters

As illustrated in Figure 12, the inverter was an AC source to create suitable power that was injected into the grid, and the DC/DC converter was a DC source to regulate the DC link capacitor. Thus, the need for a droop controller to avoid the circulating power and a DC link voltage controller to regulate the voltage [29,30].

### 5.1. DC Link Modeling

To keep the inverters working without interruption, a controller must be developed to reduce the amount of imported power. Therefore, a DC/DC converter regulates the

DC link voltage when the power flows normally from the inverter, as shown in Figure 13. The reference of the DC-link voltage is expressed by the VDC link, but when the inverter imports power, this means that the DC link exceeds the limit, and the controller must change the power references to keep the inverter working, otherwise the inverters will be interrupted, which reduces the DC link voltage [31].

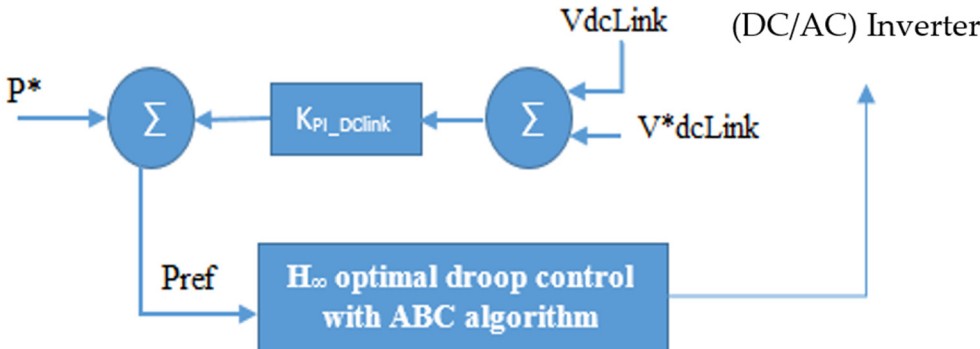

**Figure 13.** DC link voltage controller.

The frequency and voltage equations of the parallel inverters in thee microgrid by using the ABC droop control are shown in Equations (19) and (20). The external disturbances that cause an increase in the DC link voltage and inverter shutdown through the circulating current, which results from the flow of power between the distributed generators (DGs). This section aims to minimize the storage capacitor of the DC link when the DC link voltage increases. The important goal is to achieve stability between the DC link and inverter power flow by applying the H∞ PID controller with the ABC droop control to decide the setpoint power of the battery and the active and reactive reference power of AC inverters [32].

*5.2. Power Calculation and Equations of Parallel Inverters*

According to Figure 14, ABC droop control with H∞ optimal controller regulates the DC link voltage. The active power and reactive power of each inverter can be obtained as follows:

$$P_Y = \left( \frac{E_Y \, VL}{Z_o} \cos \beta_Y - \frac{V_L^2}{Z_o} \right) \cos \varphi + \left( \frac{E_Y \, VL}{Z_o} \sin \beta_Y \, \sin \varphi \right) \quad Y = 1,2 \qquad (22)$$

$$Q_Y = \left( \frac{E_Y \, VL}{Z_o} \cos \beta_Y - \frac{V_L^2}{Z_o} \right) \sin \varphi - \left( \frac{E_Y \, VL}{Z_o} \sin \beta_Y \, \cos \varphi \right) \quad Y = 1,2 \qquad (23)$$

where $Y$ is the number of the inverters; $E$ is the effective value of output voltage inverter; $\beta$ is the phase angle between the output voltage inverter and the Common AC bus; and $V_L$ is the load voltage. $Z_o$ and $\varphi$ are the equivalent output reactance and resistance of each inverter, respectively [33].

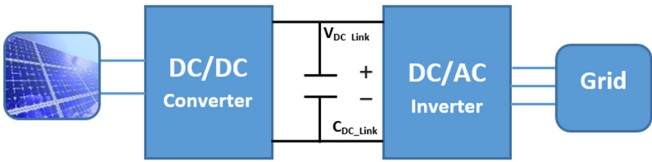

**Figure 14.** Microgrid topology.

Equations (26) and (27) are obtained by the linearity of Equations (17), (19), (20), (22) and (23), where $K_{PE}$, $K_{P\beta}$, $K_{qE}$, and $K_{q\beta}$ are the coefficients of linearity.

$$\Delta P = C * \left[\frac{\partial P}{\partial E}\right]\Delta E + C * \left[\frac{\partial P}{\partial \beta}\right]\Delta\beta \tag{24}$$

$$\Delta Q = C * \left[\frac{\partial Q}{\partial E}\right]\Delta E + C * \left[\frac{\partial Q}{\partial \beta}\right]\Delta\beta \tag{25}$$

$$\Delta P = \begin{bmatrix} K_{PE_{11}} & \cdots & K_{PE_{1n}} \\ \vdots & \ddots & \vdots \\ K_{PE_{m1}} & \cdots & K_{PE_{mn}} \end{bmatrix} * \left[\frac{\partial P}{\partial E}\right]\Delta E + \begin{bmatrix} K_{P\beta_{11}} & \cdots & K_{P\beta_{1n}} \\ \vdots & \ddots & \vdots \\ K_{P\beta_{m1}} & \cdots & K_{P\beta_{mn}} \end{bmatrix} * \left[\frac{\partial P}{\partial \beta}\right]\Delta\beta \tag{26}$$

$$\Delta Q = \begin{bmatrix} K_{qE_{11}} & \cdots & K_{qE_{1n}} \\ \vdots & \ddots & \vdots \\ K_{qE_{m1}} & \cdots & K_{qE_{mn}} \end{bmatrix} * \left[\frac{\partial Q}{\partial E}\right]\Delta E + \begin{bmatrix} K_{q\beta_{11}} & \cdots & K_{q\beta_{1n}} \\ \vdots & \ddots & \vdots \\ K_{q\beta_{m1}} & \cdots & K_{q\beta_{mn}} \end{bmatrix} * \left[\frac{\partial P}{\partial \beta}\right]\Delta\beta \tag{27}$$

According to the case study, there are two inverters. The change output powers ($\Delta P$ and $\Delta Q$) of the first and second inverters are represented in Equations (28)–(31) and the small signal model is shown in Figure 12:

$$\Delta P_1 = [A_1] * \Delta E_1 + [A_2] * \Delta E_2 + [B_1] * \Delta\beta_1 + [B_2] * \Delta\beta_2 \tag{28}$$

$$\Delta P_2 = [A_3] * \Delta E_1 + [A_4] * \Delta E_2 + [B_3] * \Delta\beta_1 + [B_4] * \Delta\beta_2 \tag{29}$$

$$\Delta Q_1 = [A_5] * \Delta E_1 + [A_6] * \Delta E_2 + [B_5] * \Delta\beta_1 + [B_6] * \Delta\beta_2 \tag{30}$$

$$\Delta Q_2 = [A_7] * \Delta E_1 + [A_8] * \Delta E_2 + [B_7] * \Delta\beta_1 + [B_8] * \Delta\beta_2 \tag{31}$$

The parameters $A_1$, $A_2$, $A_3$, $A_4$, $A_5$, $A_6$, $A_7$, $A_8$, $B_1$, $B_2$, $B_3$, $B_4$, $B_5$, $B_6$, $B_7$, and $B_8$ can be calculated as given in Appendix A. The following equations can be obtained from Equations (19) and (20).

$$\Delta\omega_{1,2} = -\begin{bmatrix} K_{P_{11}} & \cdots & K_{P_{1n}} \\ \vdots & \ddots & \vdots \\ K_{P_{m1}} & \cdots & K_{P_{mn}} \end{bmatrix} * \Delta P_{avg\ 1,2} \tag{32}$$

$$\Delta E_{1,2} = -\begin{bmatrix} K_{q_{11}} & \cdots & K_{q_{1n}} \\ \vdots & \ddots & \vdots \\ K_{q_{m1}} & \cdots & K_{q_{mn}} \end{bmatrix} * \Delta Q_{avg\ 1,2} \tag{33}$$

Calculate the average values of active power $\Delta P_{avg}$ and reactive power $\Delta Q_{avg}$ with the relation of $\Delta P$ and $\Delta Q$ as follows:

$$\Delta P_{avg1,2} = \frac{1}{\tau s + 1}\Delta P_{1,2} \tag{34}$$

$$\Delta Q_{avg\ 1,2} = \frac{1}{\tau s + 1}\Delta Q_{1,2} \tag{35}$$

where $\tau$ is a time constant.

*5.3. Design of the DC Link Voltage*

The relation between the energy absorbed ($E_{DC\_Link}$) by the DC link capacitor and the voltage drop of the link ($V_{DC\_Link}$) can be expressed as:

$$E_{DC\_Link} = \int P(t)\, dt = \frac{1}{2} C_{DC\_Link} V_{DC\_Link}^2 \tag{36}$$

where $P$ is the absorb of power instantaneous; $V_{DC\_Link}$ is the voltage drop on the link capacitor; and $C_{DC\_Link}$ is the DC link capacitor. The equations can be linearized as follows:

$$\frac{dE_{DC\_Link}}{dt} = \frac{\int P(t)\, d(t)}{dt} \tag{37}$$

$$S\,\Delta E_{1,2} = \Delta P_{1,2} \tag{38}$$

$$S\,\Delta \beta_{1,2} = \Delta \omega_{1,2} \tag{39}$$

Substitute Equation (38) into Equation (36), and represent the new equation as shown in Equation (40) to calculate the change value in the DC link voltage variation $\Delta V$.

$$\Delta V_{DC\_Link} = \frac{L_o * \Delta P}{S * C_{DC\_Link}} \tag{40}$$

where $\Delta V$ is the DC link voltage variation value, and $L_o$ is the small change value of the DC link voltage. Equations (28)–(31) are substituted in Equations (34) and (35), rewritten as follows:

$$S\,\Delta P_{avg1,2} = \frac{1}{\tau}\left(\Delta P_{1,2} - \Delta P_{avg1,2}\right) \tag{41}$$

$$S\,\Delta Q_{avg1,2} = \frac{1}{\tau}\left(\Delta Q_{1,2} - \Delta Q_{avg1,2}\right) \tag{42}$$

All equations are represented in Appendices B and C.

## 6. Simulation Results and Verification

Power-sharing between DGs and the voltage–frequency stability controller needs an optimization response with high performance. In this work, the output waveform of the current and the power-sharing analysis were obtained by the H∞ controller with the ABC algorithm. The results were compared with conventional droop control, PSO droop control, and ABC droop control under the same working conditions. The number of iterations and particles were set at 100 and 50, respectively, for the PSO and ABC algorithm. The proportional gain range was $0 < kp < 30$. The system was developed as the parameters of the PD controller $0 < kd < 10$ and $K_I = 0$. The number of onlooker bees in the ABC algorithm was equal to the population size of 100. The colony size was 100; all control parameters were imposed as $-20,000 < \theta_k < 2000$, k = {1, 2 . . . . . . , colony size}; and the weight factor of $a_i$ in Equation (14) as $a_1 = 5$, $a_2 = a_3 = a_4 = a_5 = 1$. The time T in (14) was set as 100 s, in addition to the infinity-optimized controller $P_\infty \leq 1$. Here, a case study illustrates the proposed design procedures: suppose that the microgrid consists of two inverters that have the same power rating, power-sharing will be shared evenly between the two inverters, as shown in Table 2. The effects of the algorithms were studied for the following cases:

*6.1. Active Power Sharing among Two Parallel Inverters*

Power-sharing among the parallel inverters is designed to achieve an optimal response. Several controllers are applied in the microgrid according to the case study conditions, and the most important, minimizing the overshoot that may arise while switching the microgrid after 0.02 s from grid-connected mode to island mode by a three-phase circuit breaker, while

at the same time maintaining a match of the power-sharing among the inverters. The H∞ controller combined with the ABC algorithm is considered an intelligent method, where the optimal parameters are used in the power controller and the selection of optimal parameters minimizes the error integrating fitness function and ensures the optimal transient response of the microgrid. Figure 15 illustrates the active power load sharing between inverter 1 and inverter 2 according to the case study referred to in Table 2. A load of active power was 7500 W, and the active power nominal value was 10 KW per inverter. In the conventional droop control, the active power generated by the first inverter and the second inverter was from 3300 to 4500 W and 3500 to 4540 W, respectively. In addition, the overshoot value at the grid switching to the island mode was very high. As shown in Figure 15a, the power-sharing between inverters was not accurate where the change in power was very high. In droop control with the PSO algorithm, the first and second inverter generated an average power from 3600 W to 3730 W, and the overshoot value at switching to island mode in the first inverter was 13.33% and the second inverter was 16%, as shown in Table 3. From the results, power-sharing has an oscillation value and the match of active power between the two inverters was 81.34%, as shown in Figure 15b. To minimize the errors, sharing of the active power control method should be improved in order to achieve the optimal values of coefficient droop control that effects the network stability by the ABC algorithm and H∞ controller with ABC. Figure 15c,d illustrates the ABC algorithm and H∞ optimal controller with the ABC algorithm, respectively. The inverters start sharing power in island mode after 0.02 s by sharing 7500 W of active power; loads of the grid are connected to the power system. The first and second inverters generated active power of 3750 W, when the ABC algorithm was used, the overshoot value at switching to island mode in the first inverter was 3.42%, the second inverter was 3.85%, and the match of active power between the two inverters was 98.165%, as shown in Figure 15c. In the H∞ optimal controller using the ABC algorithm, the overshoot value at switching to island mode in the first inverter was 1.34%, the second inverter was 1.86%, and the match of active power between the two inverters were 98.64%, as shown in Figure 15d and Table 3.

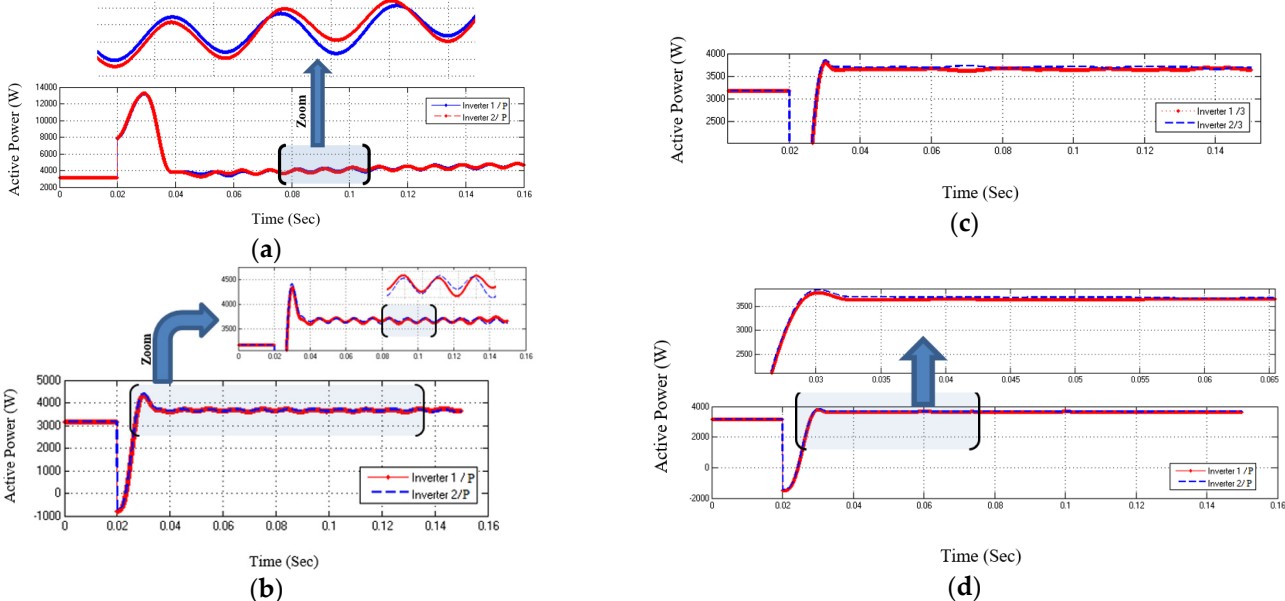

**Figure 15.** Active power sharing between inverter 1 and inverter 2 by droop control using: (**a**) conventional control; (**b**) PSO algorithm; (**c**) ABC algorithm; and (**d**) H∞ with the ABC algorithm.

**Table 3.** Comparison of power sharing with different algorithms.

| | Method | Overshoot % | | Power Matching among Inverter 1 and Inverter 2 % |
|---|---|---|---|---|
| | | $P_1$ % | $P_2$ % | |
| **DGs Active Power** | PSO Droop Control | 13.33 | 16 | 81.34 |
| | ABC Droop Control | 3.42 | 3.85 | 98.165 |
| | H∞ with ABC | 1.34 | 1.86 | 98.64 |
| | | $Q_1$ % | $Q_2$ % | |
| **DGs Reactive Power** | PSO Droop Control | 10.33 | 15.5 | 80.62 |
| | ABC Droop Control | 1.12 | 3.34 | 93.89 |
| | H∞ with ABC | 0.85 | 1.47 | 96.77 |

*6.2. Reactive Power Sharing among Two Parallel Inverters*

The load of reactive power was QL = 300 Var, and the reactive power nominal value was 150 Var per inverter. The power-sharing performance with conventional droop control at load showed the first inverter generated from −110 to 150 Var, and the second inverter generated reactive power from −100 to 150 Var, as shown in Figure 16a. Depending on the results, poor power-sharing and slow response are indicated, so the control system should be improved. In a power-sharing performance with PSO droop control, the first inverter generated an average reactive power from 180 to 150 Var, and the second inverter generated an average reactive power from 165 to 145 Var, as shown in Figure 16b, but the match of reactive power between the two inverters was 80.62% when this method was compared with the conventional droop control. The dynamic response was faster in the optimal technique, where the overshoot decreased to 10.33% in the first inverter and 15.5% in the second inverter. To minimize the error sharing of the reactive power, the control method should be improved to achieve optimal values of coefficient droop control that affects the network stability by the ABC algorithm and H∞ controller with ABC. Figure 16c,d illustrates the ABC algorithm and the H∞ optimal controller with the ABC algorithm, respectively. The inverters started sharing reactive power by sharing 300 Var and the first and second inverter generated reactive power of 150 Var when the ABC algorithm used the overshoot value at switching to island mode in the first inverter of 1.12%, the second inverter of 3.34%, and the match of reactive power between the two inverters was 93.89%, as shown in Figure 16c. In the H∞ controller with the ABC algorithm, the overshoot value at switching to island mode in the first inverter was 0.85%, the second inverter was 1.47%, and the match of active power between the two inverters was 96.77%, as shown in Figure 16d and Table 3.

*6.3. Result of the DC Link Controller*

The simulation results in Figure 15 show that the rated power of the two inverters was 10 KW. In case 1, as shown in Table 2, the load active power was 7500 W, where the output power of the first inverter and the second generated 3750 W. In case 2, Figure 17 shows the test of the DC link controller where the load active power (PL = 0) and the inverters are ready to share power, but the PL = 0, so the output power of the two inverters will be decreased to zero. The DC link voltage of the two inverters after activating the controller will be 392 V, so the two parallel inverters have a voltage Vo = 226 Vrms. This means that the effect of the DC link controller on the microgrid will prevent the voltage increase exceeding the limitation voltage, which makes the inverters work in normal situations, waiting for the updated feedback signal to decide the output power-sharing, while at the same time, keep the inverters' voltage value as constant.

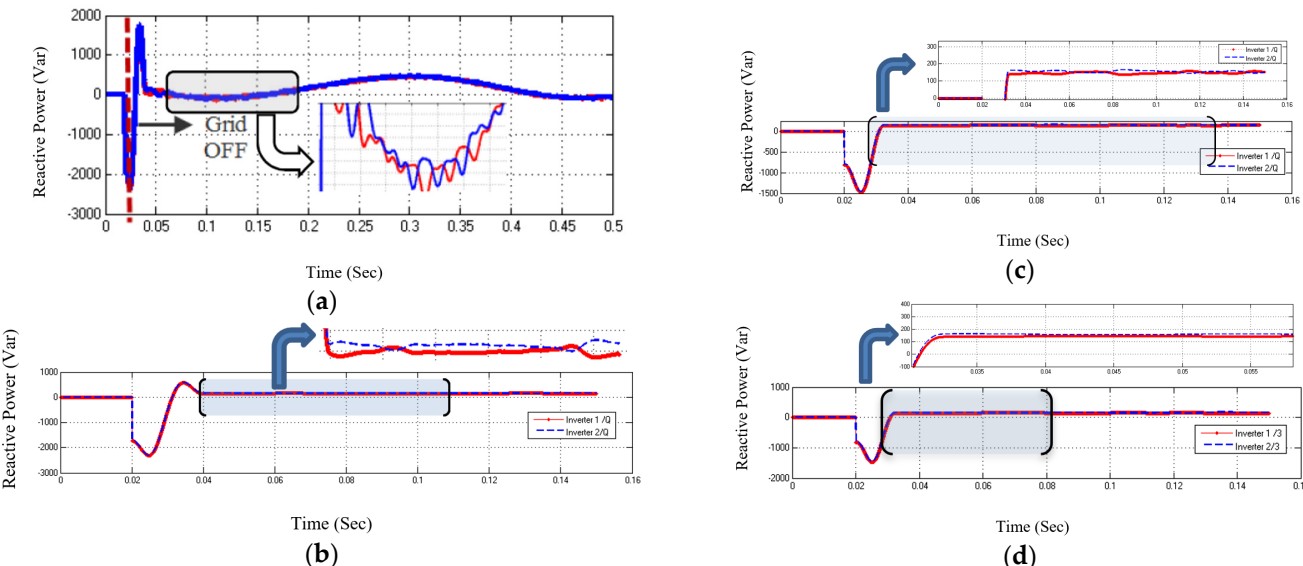

**Figure 16.** Reactive power sharing between inverter 1 and inverter 2 by droop control using: (**a**) Conventional control; (**b**) PSO algorithm; (**c**) ABC algorithm; and (**d**) H∞ with ABC.

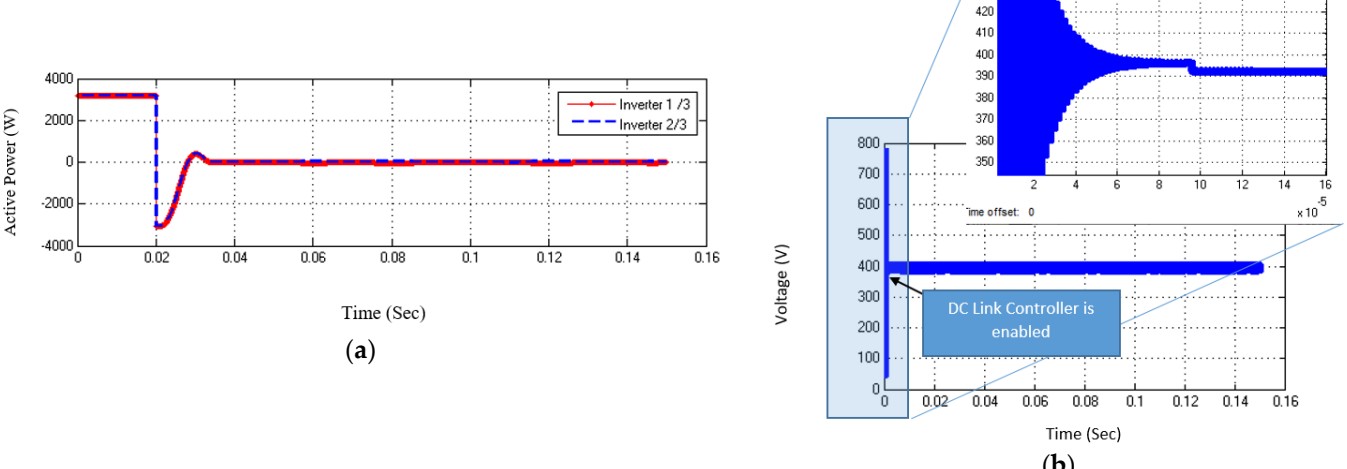

**Figure 17.** (**a**) Active power of both inverters at $P_L = 0$. (**b**) DC link voltage of inverter with the DC link controller.

*6.4. Power Sharing Signal Analysis under Variable Loads*

In this section, a case study illustrates the proposed design procedures. Suppose that a microgrid consisting of two inverters has the same power rating, and that power-sharing will be shared evenly between two inverters, as shown in Table 4.

MATLAB/Simulink was used to create a microgrid model with two inverters, where each inverter is represented as an ideal voltage source with a series inductive output impedance. The two inverters had identical parameters. This section of the simulation was used to test the proposed controller (PSO and ABC algorithm) under various load situations and compare it to the classic droop controller's performance. Figure 18 depicts the output power of the two inverters using conventional droop control under various load circumstances including low, medium, and high.

**Table 4.** Simulation parameters.

| Symbol | Description | Value |
|---|---|---|
| $P_{Max}$ | Maximum value of active power for each inverter | 13 KW |
| $Q_{Max}$ | Maximum value of reactive power for each inverter | 3.5 KW |
| $P_{L\ Low}$ | Low load active power | 7 KW |
| $Q_{L\ Low}$ | Low load reactive power | 300 Var |
| $P_{L\ Medium}$ | Medium load active power | 20 KW |
| $Q_{L\ Medium}$ | Medium load reactive power | 2 KVar |
| $P_{L\ High}$ | High load active power | 25 KW |
| $Q_{L\ High}$ | High load reactive power | 3 KVar |
| $E^*$ | Nominal peak voltage | 325 V |
| $f^*$ | Nominal frequency | 50 HZ |
| $V_{DC\_Link}$ | DC link voltage | 400 V |

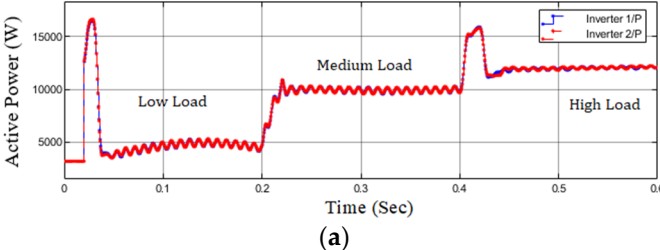 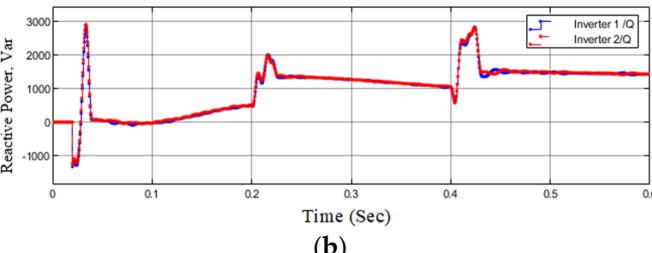

(**a**)            (**b**)

**Figure 18.** Output power of the inverter when low, medium, and high loads are supplied using conventional droop control for: (**a**) Active power; (**b**) Reactive power.

The low load at $t$ = 0.01 s to $t$ = 0.2 s as $P$ = 7 KW, and $Q$ = 300 Var. A medium load starts at $t$ = 0.2 s to $t$ = 0.4 s as $P$ = 20 KW, and $Q$ = 2 KVar. A high load starts at $t$ = 0.4 s to $t$ = 0.6 s as $P$ = 25 KW and $Q$ = 3 KVar.

Figure 18 illustrates the active and reactive power with conventional droop control under different load conditions. The maximum value of active power and reactive power were 13 KW, and 1.5 KVar, respectively, per inverter. In the conventional droop control, the first and second inverters generated oscillation output power, in addition, the overshoot value at switching from one load to another is very high. From the results, sharing power is not accurate between inverters where the rate change of power is high, as shown in Figure 18.

Figure 19 shows the droop control with the PSO algorithm, where the controller is activated by a sudden change in the active and reactive power, the load changes from low (7 KW, 300 Var) to medium (20 KW, 2 KVar) load, and after that, the load changes from medium (20 kW, 2 KVar) to high (25 kW, 3 KVar) load. During DG injection and sudden change in the load situations, the controller is programmed to set power levels with minimal settling time and overshoot. In the PSO algorithm, the out power is accurate, but the results showed an error value of the overshoot when the load changed suddenly. To minimize the error sharing of the active and reactive power, the control method should be improved in order to achieve optimal values of coefficient droop control that affects the network stability by the ABC algorithm.

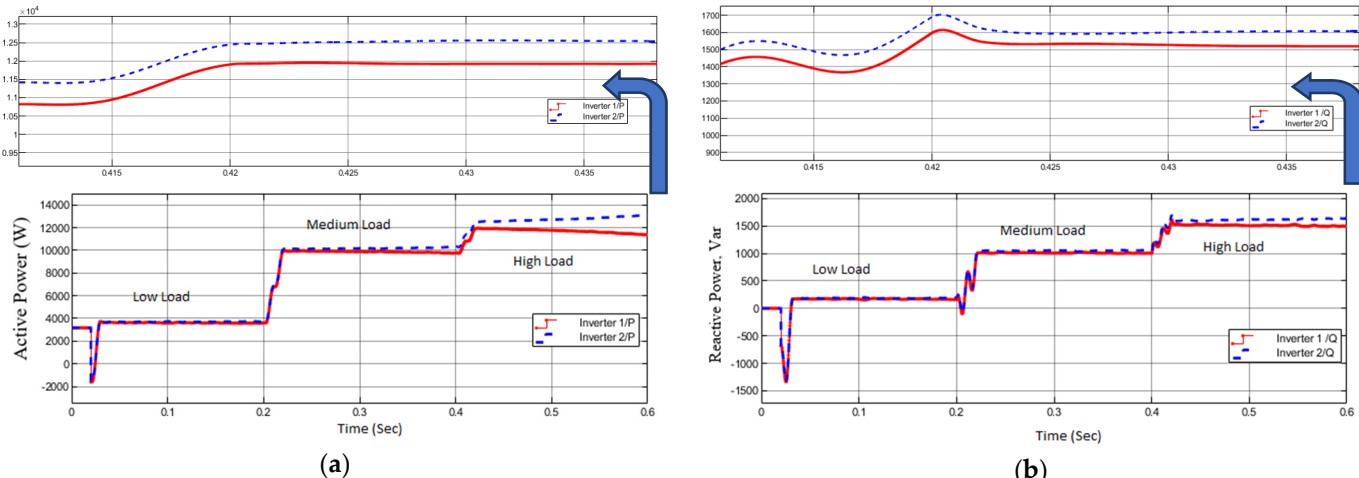

**Figure 19.** Output power of the inverter when low, medium, and high loads are supplied using PSO droop control for: (**a**) Active power; (**b**) Reactive power.

Figure 20 shows the droop control with ABC algorithm, where the controller is activated by a sudden change in the active and reactive power, the load changes from low (7 KW, 300 Var) to medium (20 KW, 2 KVar) load, and after that, the load changes from medium (20 kW, 2 KVar) to high (25 kW, 3 KVar) load. The inverters start sharing the active power in island mode after 0.01 s by sharing 7 KW of active power (3.5 KW from a first inverter, and 3.5 KW from a second one). In the medium load, the two inverters share 20 KW (10 KW from the first inverter, and 10 KW from the second one). The inverters started sharing the power in a high load by sharing 25 KW of active power (12.5 KW from the first inverter, and 12.5 KW from the second one), as shown in Figure 20a. The inverters started sharing the reactive power in island mode after 0.01 s by sharing 300 Var of reactive power (150 Var from the first inverter, and 150 Var from the second one). In the medium load, the two inverters share 2 KVar (1 KVar from the first inverter, and 1 KVar from the second one). The inverters started sharing power in the high load by sharing 3 KVar of reactive power (1.5 KVar from the first inverter, and 1.5 KVar from the second one), as shown in Figure 20b.

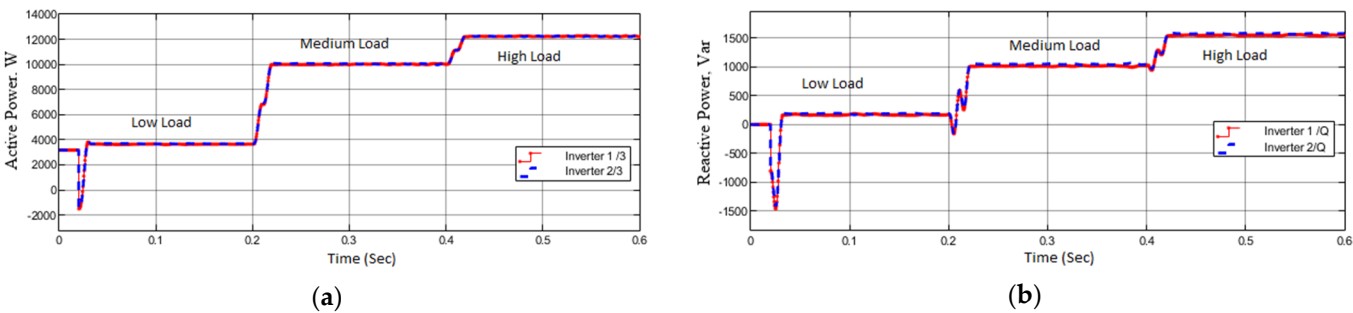

**Figure 20.** Output power of the inverter when low, medium, and high loads are supplied using ABC droop control for: (**a**) Active power; (**b**) Reactive power.

## 7. Conclusions

This article has been proposed to improve the power-sharing among parallel inverters by minimizing the overshoot response and fast dynamic response, in addition to minimizing the error cost function. The strategic objective was to optimize the droop control with the H∞ controller with the ABC algorithm, and a robust performance analysis was obtained by this method. These results were compared with the conventional droop control, PSO algorithm droop control, and ABC algorithm droop control. Thus, the simulations prove the improvement and accuracy of the design to achieve match power-sharing between the

parallel inverters and robust performance against external disturbance. A power-sharing method was presented in this article to improve active and reactive power-sharing across parallel inverters in island mode. To enhance reactive power-sharing, the suggested method employs intermittent monitoring of the PCC voltage. It is feasible to determine the value of the inverters' output impedance (inductive impedance) as well as that of the cables under these conditions. When the PCC voltage measurement is unavailable, the new estimated impedance values are utilized to compute a new value to gain a conventional droop controller that takes over control of reactive power-sharing. The new droop arrangement (H-infinity controller with artificial bee colony algorithm) enhances the active and reactive power-sharing without requiring continuous monitoring of the PCC voltage.

In particular, the main limitation of the proposed function is the limitation of the droop controller, which is used in island mode to achieve power-sharing between parallel-operated inverters. Therefore, future work will be leading with the same method that may be used to construct the multi-inverter model produced in the paper to explore the issue of island mode and transient fluctuations as a function of the number of integrated units and power set-point variances. Moreover, the hybridization between AC and DC microgrids throughout both modes of operation as well as their impact on power flow between all units may be investigated. This endeavor will expose new modeling and actual work obstacles.

**Author Contributions:** Conceptualization, M.S.J. and N.K.; Methodology, M.S.J. and N.K.; Software, M.S.J.; Validation, M.S.J. and N.K.; Formal analysis, N.K.; Investigation, N.K.; Resources, M.S.J. and N.K.; Data curation, M.S.J. and N.K.; Writing—original draft preparation, M.S.J. and N.K.; Writing—review and editing, N.K.; Visualization, M.S.J. and N.K.; Supervision, N.K.; Project administration, N.K.; Funding acquisition, M.S.J. and N.K. All authors have read and agreed to the published version of the manuscript.

**Funding:** This research was funded by "Presidency For Turks Abroad and Related Communities-YTB", grant number 15PS000120, Istanbul, Turkey.

**Data Availability Statement:** The data presented in this study are available on request from the corresponding author.

**Acknowledgments:** The authors would like to acknowledge the impressive support provided by the Turkish Scholarship to fund the research. The authors would also like to extend their thanks and appreciation to the efforts of Yildiz Technical University in support of this study.

**Conflicts of Interest:** The authors declare no conflict of interest.

## Nomenclature

| | |
|---|---|
| $P^*$ and $Q^*$ | Active and reactive power nominal value |
| $P$ and $Q$ | Active and reactive power actual value |
| $Kp$ and $kq$ | Frequency and voltage droop control gain compensator |
| $w^*$ and $E^*$ | Frequency and voltage nominal value |
| $w$ and $E$ | Frequency and voltage actual value |
| $V_i^k$ | The $ith$ particle velocity at $kth$ iteration. |
| $X_i^k$ | The particle's current position. The $i$ and $k$ are particle index, and iteration index. |
| $Pbest$ and $Pgbest$ | The particle's best position and global group's global best position. |
| $r_1$ and $r_2$ | Random numbers between {0, 1} |
| $V_{ij}$ | A new food source ($i, j$ are random selected index) |
| $X_{kj}$ | Selected random food source, and k = {1, 2, . . . , colony size (population)} |
| $\varphi_{ij}$ | A random number $\{-1, 1\}$ |
| $G(s)$ | Uncertain plant of the closed-loop system |
| $Yp(t)$, and $R(t)$ | The output system, and a reference input |
| $e(t)$, and $C(s)$ | An error signal of the closed-loop control system, and the tuning algorithm of PID controller |

| | |
|---|---|
| *Wd*, and *Wn* | Weighting functions |
| $D(t)$ and $N(t)$ | The disturbance and the noise |
| $T(s)$ and $D(s)$ | Complementary function, and sensitivity function |
| $P\infty$, $\tau$ | The robust stability performed by H∞ optimization, and a time constant. |
| OS, $t_r$, $t_s$, ESS, ai | Overshoot, rising time, settling time, steady-state error, weight factor ($i = 1, 2 \ldots, 5$) |
| $I(\theta)$ | The performance for weighting function, $\theta$ is the coefficient controller |
| $K_{PID}$ | $K_P$, $K_I$, and $K_D$ are proportional, integral, and derivative controllers |
| $\beta$ | The phase angle between the output voltage inverter and the Common AC bus |
| $V_L$, $Z_o$, $\varphi$ | Load voltage, the equivalent output reactance and resistance of each inverter |
| *KPE*, *KPβ*, *KqE*, *Kqβ* | The coefficients of linearity |
| $\Delta P$ and $\Delta Q$ | The change output active power and reactive power |
| $\Delta Pavg$ and $\Delta Qavg$ | Average values of the change active power and reactive power |
| $E_{DC\_Link}$ | The energy absorbed by the DC link capacitor |
| $V_{DC\_Link}$ | The voltage drops on the link capacitor |
| $C_{DC\_Link}$ | The DC link capacitor |
| $\Delta V_{DC\_Link}$ | The DC link voltage variation value |

## Appendix A. The Change Output Powers of the First and Second Inverters Coefficients

The parameters $A_1$, $A_2$, $A_3$, $A_4$, $A_5$, $A_6$, $A_7$, $A_8$, $B_1$, $B_2$, $B_3$, $B_4$, $B_5$, $B_6$, $B_7$, and $B_8$ can be calculated as follows:

$$A_1 = \frac{\partial P_1}{\partial E_1} = \left( \frac{VL}{Z_o} \cos \beta_1 \right) \cos \varphi + \left( \frac{VL}{Z_o} \sin \beta_1 \right) \sin \varphi \tag{A1}$$

$$A_2 = \frac{\partial P_1}{\partial E_2} = 0 \tag{A2}$$

$$B_1 = \frac{\partial P_1}{\partial \beta_1} = \left( \frac{E_1 \, VL}{Z_o} \cos \beta_1 \right) \sin \varphi - \left( \frac{E_1 \, VL}{Z_o} \sin \beta_1 \right) \cos \varphi \tag{A3}$$

$$B_2 = \frac{\partial P_1}{\partial \beta_2} = 0 \tag{A4}$$

$$A_3 = \frac{\partial P_2}{\partial E_1} = 0 \tag{A5}$$

$$A_4 = \frac{\partial P_2}{\partial E_2} = \left( \frac{VL}{Z_o} \cos \beta_2 \right) \cos \varphi + \left( \frac{VL}{Z_o} \sin \beta_2 \right) \sin \varphi \tag{A6}$$

$$B_3 = \frac{\partial P_2}{\partial \beta_1} = 0 \tag{A7}$$

$$B_4 = \frac{\partial P_2}{\partial \beta_2} = \left( \frac{E_2 \, VL}{Z_o} \cos \beta_2 \right) \sin \varphi - \left( \frac{E_2 \, VL}{Z_o} \sin \beta_2 \right) \cos \varphi \tag{A8}$$

$$A_5 = \frac{\partial Q_1}{\partial E_1} = \left( \frac{VL}{Z_o} \cos \beta_1 \right) \sin \varphi - \left( \frac{VL}{Z_o} \sin \beta_1 \right) \cos \varphi \tag{A9}$$

$$A_6 = \frac{\partial Q_1}{\partial E_2} = 0 \tag{A10}$$

$$B_5 = \frac{\partial Q_1}{\partial \beta_1} = -\left( \frac{E_1 \, VL}{Z_o} \cos \beta_1 \right) \cos \varphi - \left( \frac{E_1 \, VL}{Z_o} \sin \beta_1 \right) \sin \varphi \tag{A11}$$

$$B_6 = \frac{\partial Q_1}{\partial \beta_2} = 0 \tag{A12}$$

$$A_7 = \frac{\partial Q_2}{\partial E_1} = 0 \tag{A13}$$

$$A_8 = \frac{\partial Q_2}{\partial E_2} = \left(\frac{\text{VL}}{\text{Z}_\text{o}} \cos \beta_2\right) \sin \varphi - \left(\frac{\text{VL}}{\text{Z}_\text{o}} \sin \beta_2\right) \cos \varphi \tag{A14}$$

$$B_7 = \frac{\partial Q_2}{\partial \beta_1} = 0 \tag{A15}$$

$$B_8 = \frac{\partial Q_2}{\partial \beta_2} = -\left(\frac{E_2 \, \text{VL}}{\text{Z}_\text{o}} \cos \beta_2\right) \cos \varphi - \left(\frac{E_2 \, \text{VL}}{\text{Z}_\text{o}} \sin \beta_2\right) \sin \varphi \tag{A16}$$

**Appendix B**

The average values of active $\Delta P_{avg}$ and reactive $\Delta Q_{avg}$ power are represented as follows:

$$S \, \Delta P_{avg1} = \frac{1}{\tau}([A_1] * \Delta E_1 + [A_2] * \Delta E_2 + [B_1] * \Delta \beta_1 + [B_2] * \Delta \beta_2) - \frac{1}{\tau}\Delta P_{avg1} \tag{A17}$$

$$S \, \Delta P_{avg2} = \frac{1}{\tau}([A_3] * \Delta E_1 + [A_4] * \Delta E_2 + [B_3] * \Delta \beta_1 + [B_4] * \Delta \beta_2) - \frac{1}{\tau}\Delta P_{avg2} \tag{A18}$$

$$S \, \Delta Q_{avg1} = \frac{1}{\tau}([A_5] * \Delta E_1 + [A_6] * \Delta E_2 + [B_5] * \Delta \beta_1 + [B_6] * \Delta \beta_2) - \frac{1}{\tau}\Delta Q_{avg1} \tag{A19}$$

$$S \, \Delta Q_{avg2} = \frac{1}{\tau}([A_7] * \Delta E_1 + [A_8] * \Delta E_2 + [B_7] * \Delta \beta_1 + [B_8] * \Delta \beta_2) - \frac{1}{\tau}\Delta Q_{avg2} \tag{A20}$$

**Appendix C**

The state-space model can be illustrated as shown in (A21):

$$S \begin{bmatrix} \Delta\beta_1 \\ \Delta\beta_2 \\ \Delta E_1 \\ \Delta E_2 \\ \Delta\omega_1 \\ \Delta\omega_2 \\ \Delta P_{avg1} \\ \Delta P_{avg2} \\ \Delta Q_{avg1} \\ \Delta Q_{avg2} \\ \Delta V_{DC_{Link1}} \\ \Delta V_{DC_{Link2}} \end{bmatrix} = \begin{bmatrix} 0 & 0 & 0 & 0 & 1 & 0 & 0 & 0 & 0 & 0 \\ 0 & 0 & 0 & 0 & 0 & 1 & 0 & 0 & 0 & 0 \\ -\frac{K_p}{\tau}\beta_1 & -\frac{K_p}{\tau}\beta_2 & -\frac{K_p}{\tau}A_1 & -\frac{K_p}{\tau}A_2 & 0 & 0 & \frac{K_p}{\tau} & 0 & 0 & 0 \\ -\frac{K_p}{\tau}\beta_3 & -\frac{K_p}{\tau}\beta_4 & -\frac{K_p}{\tau}A_3 & -\frac{K_p}{\tau}A_4 & 0 & 0 & 0 & \frac{K_p}{\tau} & 0 & 0 \\ -\frac{K_p}{\tau}\beta_5 & -\frac{K_p}{\tau}\beta_6 & -\frac{K_p}{\tau}A_5 & -\frac{K_p}{\tau}A_6 & 0 & 0 & 0 & 0 & \frac{K_q}{\tau} & 0 \\ -\frac{K_p}{\tau}\beta_7 & -\frac{K_p}{\tau}\beta_8 & -\frac{K_p}{\tau}A_7 & -\frac{K_p}{\tau}A_8 & 0 & 0 & 0 & 0 & 0 & \frac{K_q}{\tau} \\ \frac{\beta_1}{\tau} & \frac{\beta_2}{\tau} & \frac{A_1}{\tau} & \frac{A_2}{\tau} & 0 & 0 & -\frac{1}{\tau} & 0 & 0 & 0 \\ \frac{\beta_3}{\tau} & \frac{\beta_4}{\tau} & \frac{A_3}{\tau} & \frac{A_4}{\tau} & 0 & 0 & 0 & -\frac{1}{\tau} & 0 & 0 \\ \frac{\beta_5}{\tau} & \frac{\beta_6}{\tau} & \frac{A_5}{\tau} & \frac{A_6}{\tau} & 0 & 0 & 0 & 0 & -\frac{1}{\tau} & 0 \\ \frac{\beta_7}{\tau} & \frac{\beta_8}{\tau} & \frac{A_7}{\tau} & \frac{A_8}{\tau} & 0 & 0 & 0 & 0 & 0 & -\frac{1}{\tau} \\ \frac{L_o\,\beta_1}{C} & \frac{L_o\,\beta_2}{C} & \frac{L_o\,A_1}{C} & \frac{L_o\,A_2}{C} & 0 & 0 & 0 & 0 & 0 & 0 \\ \frac{L_o\,\beta_3}{C} & \frac{L_o\,\beta_4}{C} & \frac{L_o\,A_3}{C} & \frac{L_o\,A_4}{C} & 0 & 0 & 0 & 0 & 0 & 0 \end{bmatrix} \begin{bmatrix} \Delta\beta_1 \\ \Delta\beta_2 \\ \Delta E_1 \\ \Delta E_2 \\ \Delta\omega_1 \\ \Delta\omega_2 \\ \Delta P_{avg1} \\ \Delta P_{avg2} \\ \Delta Q_{avg1} \\ \Delta Q_{avg2} \\ \Delta V_{DC\_Link1} \\ \Delta V_{DC\_Link2} \end{bmatrix} \tag{A21}$$

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
