# Peer review of "Improved Optimal Control of Transient Power Sharing in Microgrid Using H-Infinity Controller with Artificial Bee Colony Algorithm"

_energies, doi:10.3390/en15031043_

Round 1
Reviewer 1 Report
The paper presents actual topic, that can be interesting to a wide audience. However, some parts can be improved:
- It is unusual to put the figures in the Introduction, it can be moved to Section 2.
- The authors explained the objectives of the paper, but it is not noticeable enough, it should be separated into a new line from This paper proposes...
- At the beginning of Section 5 is stated Figure 1 shows ... Should it be Figure 9?
- Finally, the authors could be more critical in the conclusion, ie point out possible shortcomings and limitations of their research and give a hint in which direction their work could continue in the future.
Author Response
Dear Reviewer,
Please see the attachment.
Best Regards,
Mohammed Said Jouda

Reviewer 2 Report
Energies Jouda
Comments to the Editor
The paper presents a novel controller based on the Bee Colony Algorithm for the optimal power sharing in microgrids.
All in all, my opinion is that the manuscript cannot be published in its current form and it requires a major revision.
Comments to the authors
The paper presents a novel controller based on the Bee Colony Algorithm for the optimal power sharing in microgrids.
All in all, my opinion is that the manuscript cannot be published in its current form and it requires a major revision.
All the following indicated aspects should be clarified and better explained in the manuscript.
Literature review
- The main contributions of the paper are clearly described. Nevertheless, from the current manuscript it is not grasp understanding the novelty of the work. The authors should better highlight the innovative aspects of their work in the manuscript.
System design
- The description of the proposed methodology could be improved. First, it could be better to insert at the beginning of Section II an outline about the system scheme/architecture (how many components, the aim of each components, the actors involved in each step, etc.); here, a high-level diagram/scheme could also help reader following the whole design/validation description.
- The authors should motivate the choice of the Bee Colony algorithm as computational method used to solve the addressed optimization problem. However, several recent scientific studies on energy management of microgrids, show that HHO and WCA performs better than other advanced algorithms. The Authors should comment this point.
- M. Helmi et al., "Efficient and Sustainable Reconfiguration of Distribution Networks via Metaheuristic Optimization," in IEEE Transactions on Automation Science and Engineering, vol.19, issue.1, 2022.
- A. Muhammad et al., "Distribution Network Planning Enhancement via Network Reconfiguration and DG Integration Using Dataset Approach and Water Cycle Algorithm," in Journal of Modern Power Systems and Clean Energy, vol. 8, no. 1, pp. 86-93, January 2020.
(documents that could be cited in the text).
Problem formulation
- The authors should clearly characterize the overall problem that they intend to solve. What type of decision variables (i.e. integer, real, etc) and how many? How many constraints (bounding, inequality, equality)?
Case study
- Is the case study based on real data? How are the data generated?
- The author should report the running time of simulations (see the comment on large-scale dimensionality).
Conclusions
- Conclusions needs to be extended to present further implications for future research and many managerial insights based on the results of the study, as well as limitations.
Minor
- The authors should check that all the used acronyms are explained the first time they are used.
- Mainly the English is good and there are only a few typos. However the paper should be carefully rechecked.
Author Response

(The authors gave the same response as above.)

Round 2
Reviewer 2 Report
In the revised paper several improvements have been added.
Previous comments and concerns have been sufficiently addressed, except the following ones (corresponding to point 3. of the previous review report), whose discussion has not been included neither in conclusions or in model assumptions, requiring to be corroborated by references.
The comparison with the conventional PID is clear. However, the authors should motivate the choice of the Bee Colony algorithm as computational method used to solve the addressed optimization problem. However, several recent scientific studies on energy management of microgrids, show that HHO and WCA performs better than other traditional metaheuristics algorithms.
- M. Helmi et al., "Efficient and Sustainable Reconfiguration of Distribution Networks via Metaheuristic Optimization," in IEEE Transactions on Automation Science and Engineering, vol.19, issue.1, 2022.
- A. Muhammad et al., "Distribution Network Planning Enhancement via Network Reconfiguration and DG Integration Using Dataset Approach and Water Cycle Algorithm," in Journal of Modern Power Systems and Clean Energy, vol. 8, no. 1, pp. 86-93, January 2020.
(documents that could be cited in the text).
Author Response

(The authors gave the same response as above.)
